# Anti-Interference Spectral Confocal Sensors Based on Line Spot

**DOI:** 10.3390/s25051337

**Published:** 2025-02-22

**Authors:** Bo Wang, Jiafu Li, Mingzhe Luo, Fengshuang Liang, Jiacheng Hu

**Affiliations:** 1Key Laboratory of In-Situ Metrology, Ministry of Education, China Jiliang University, Hangzhou 310018, China; wb.forever@aliyun.com (B.W.); hujiacheng@cjlu.edu.cn (J.H.); 2Precision Measurement Laboratory, National Institute of Metrology, Beijing 100029, China; tim123255@126.com; 3School of Mechanical and Power Engineering, Harbin University of Science and Technology, Harbin 150080, China; 2310103014@stu.hrbust.edu.cn

**Keywords:** spectral confocal, line spot, surface profile measurement, stability, anti-interference

## Abstract

Spectral confocal displacement sensors are non-contact optoelectronic sensors widely utilized for their high accuracy, speed, and ability to measure diverse surfaces. However, challenges including vibration, angular deflection, and surface quality variations can reduce sensor stability and accuracy when performing measurements such as lithium battery wafer thickness, wafer warpage, and optical component surface topography. This study proposes a line-spot-based measurement method using a binary diffractive lens and cylindrical lens with a 20× objective, and then the overall structure is simulated and optimized by using ZEMAX, which realizes a confocal measurement system with a measurement range of 800 μm, line spot length of 3.8 mm, and width of 0.2 mm. The system, calibrated with a nanometer displacement stage, achieved 30 nm resolution and significantly improved dynamic stability (standard deviation (SD) of 0.013 μm) compared to a point spectral confocal sensor (SD of 0.064 μm). The results indicate the proposed sensor exhibits improved stability during scanning measurements.

## 1. Introduction

Spectral confocal measurement technology is a critical component in non-contact measurement systems owing to its high precision, real-time capability, non-destructive nature, high axial resolution [1,2,3,4], and compatibility with diverse materials. It has broad applications in precision measurement, industrial inspection, film thickness measurement, biomedicine, and integrated circuits [5,6,7,8,9,10]. Spectral confocal sensors are primarily used for displacement and thickness measurements and can be integrated with two-dimensional motion platforms to perform surface topography and roughness measurements. As their application scenarios expand, research on this technology has intensified.

In signal processing and correction, Wang [11] uses the nonlinear mapping ability of LSTM neural networks to realize the direct characterization from the full-spectrum signal to the position information, extract the peak wavelength more accurately, reduce the peak wavelength and position of the polynomial fitting calibration of the existence of nonlinear errors, and improve the accuracy and resolution of the measurement. Wang [12] proposed an adaptive correction method to address the low signal-to-noise ratio or distortion of the peak signals in the acquired images. This method enhances image quality through decomposing, filtering, and Gaussian enhancement, resulting in high-contrast, uniform image, and improved dynamic measurement range and sensor adaptability. Based on the Rayleigh–Sommerfeld diffraction and dispersion theories, Xi [13] studied the reflection characteristics of incident light and developed a color confocal light intensity response model that considered the incident angle, with the measurement error reduced by 72%. Additionally, Zhang [14] constructed an adaptive error correction system with translation and tilt correction algorithms, enabling accurate lens thickness measurement. These algorithms also compensate for measurement errors caused by mechanical inaccuracy, improving lens measurement accuracy by 75.21% and 71.48%, respectively.

Regarding lens optimization, to increase the measurement range of the sensor, Yang [15] proposed a method combining axial chromatic aberration theory with optical design software to develop a lens with 0.8 μm axial resolution and 3.9 mm dispersion range. Similarly, Liao [16] integrated the advantages of simple uniaxial structures with the high light flux of biaxial structures and designed an off-axis co-optical path system, achieving a sensor scanning line length of 12.5 mm and an axial measurement range exceeding 20 mm in the 450–750 nm wavelength range. Liu [17] utilized a phase Fresnel zone plate (FZP) as a dispersive objective lens, utilizing its strong dispersive capability to achieve an axial resolution of 0.8 μm and a measurement range exceeding 16 mm confocal sensor. He [18] constructed a compact long-axis dispersive objective lens using six lenses with a dispersion range of 12 mm.

The linearity of the chromatic focal shift curve affects the high-speed measurement, accuracy, and axial resolution of the sensor. To optimize this curve, Zhou [19] designed a polarization-folded, ultra-broadband, line-spectral confocal displacement sensor using a binary lens. A polarization-folded lens group corrected the chromatic focal shift curve of the binary lens, ensuring strict linearity in the axial dispersion curve. Simulation and analysis showed that the sensor achieved a linearity coefficient (R2) of 1 and a system resolution of 0.1 μm. To improve the system’s imaging quality and higher resolution, Sun [20] proposed a fiber chromatic confocal method with a tilt-coupled light source module, which reduces the width of the confocal axial response characteristic curve at each wavelength and enhances the intensity contrast of the spectral signals by adjusting the tilt-coupling angle between the optical fiber and the light source module, which ultimately results in a reduction in the full width at half maximum (FWHM) by about 17% and an improvement of the axial resolution by 1.7 times. Li [21] achieves an axial dispersion of 1215 µm in the wavelength range of 420–620 nm with a linear correlation coefficient (R2) of 0.9969 and a resolution of 6.075 nm using a radial GRIN lens, which provides superior dispersion linearity compared to conventional dispersive objective lenses.

For structural optimization, Falak [22] designed a set of compact and simple lenses using the natural chromatic aberration of hyperbolic superlenses, achieving a measurement standard deviation of 1.37 µm across a 227 µm range. Additionally, Bai [23] incorporated an annular aperture into the dispersion probe, enabling an annular beam to maintain a consistent angle of ejection and incidence. This adjustment reduces the effect of the normal direction tilting away from the optical axis of the dispersion probe, thus reducing measurement error. Chen [24] proposed a quasi-regional scanning color confocal microscope for full-field surface profile measurement. This system integrates optical line-scan illumination, imaging modules, and an electric galvanometer to achieve line scanning. These modifications alter the traditional scanning approach, reduce the influence of two-dimensional platform motion, and pair with a double telecentric design to enable high-speed inspection with a large range and high accuracy.

Regarding source selection, Shimizu et al. introduced a mode-locked femtosecond laser source in Japan, characterized by high spatial coherence and stability for dispersive confocal measurements. The intensity ratio of the two spectral signals at the conjugate focusing and defocusing positions was used as the axial response of the confocal system [25], addressing the uneven spectral distribution in mode-locked femtosecond laser sources. Sato [26] used the wavelength at the intersection of the two spectral signal curves to achieve displacement decoding, achieving an axial resolution of 20 nm within a 50 μm range. Chen [27] extended the measurement of the primary and secondary flaps to 250 μm using the axial response from the normalized and inverse normalized intensity ratios. Sato [28] improved the thermal stability of the confocal system using low-thermal-expansion materials and shortened the optical path length of the laser beam.

In terms of stability improvement, in Zhao [29], in order to reduce the uncertainty of measurement, based on CCSI, a signal processing method based on Hilbert-Huang was proposed. Compared with Fourier transform and wavelet transform, this method has better stability, noise resistance, and effectiveness. Dai [30], in order to solve the problem that the peak features of the spectral power distribution signals will be significantly attenuated with the change in scanning mode, proposes a method of obtaining the two peak features with smaller half-peak full widths by shifting the filter slit, and this method makes the feature localization more robust. Zhang [31] proposed a measurement method based on multi-sensor and multi-combination, combining color confocal sensor and eddy current sensor for measurement. Compared with the measurement based on terahertz and acoustic wave, this method has the advantages of strong anti-interference and high precision.

Overall, the current research on spectral confocal displacement sensors mainly focused on signal processing and optimization, range expansion, resolution enhancement, error reduction, and wide-field lens design for complete surface measurements. However, there is little research on the impact of interference during measurement, especially in surface profile scanning, and the primary focus is still on improving the overall system stability from the aspect of signal extraction. In this paper, we mainly optimize the structure and propose a spectral confocal sensor with a line spot to reduce the influence of vibration and other disturbing factors during the surface profile scanning measurement process and improve the overall stability of the system.

## 2. Principles and Structural Design

### 2.1. Measurement Principle

The measurement principle of the traditional point spot spectral confocal system is illustrated in Figure 1a; it primarily utilizes the principle of optical dispersion, where a binary diffractive lens decomposes white light along the optical axis into a continuous spectral range. When a reflective object is positioned within this continuous spectral range along the optical axis, the light focused on the object’s surface is reflected through a Y-shaped optical fiber and transmitted to the spectrometer owing to the reversibility of the light. The spectrometer collects the wavelength signals of the reflected light, and a nanometer displacement stage is then used to encode the collected signals, generating a wavelength-displacement calibration curve. During the actual measurements, the collected signals were decoded using a pre-calibrated curve to obtain accurate displacement values. The principle of the line spot spectral confocal system measurement is illustrated in Figure 1b; in order to realize a large spot for the purpose of improving stability, we have improved the overall structure based on the traditional point spot by adding a cylindrical mirror and a microscope objective. Secondly, the signal captured by the improved system is also different in that the signal captured is a superposition of all the point signals within the surface scanned by the line spot, and then the peak wavelength is extracted. This peak signal is determined by a portion of the characteristic points on the line spot, which are described in the dotted box on the right side of Figure 1b, depending on the intersection of the dispersion of different wavelengths with the measured surface, and the wavelength with the most intersecting points is my peak wavelength. The surface depth information is analyzed according to the peak wavelength, and the depth information obtained is used as the measurement value of the area swept by the spot, which is equivalent to a filtering process on the original signal.

### 2.2. Overall Structural Design of Line Spot Spectral Confocal Sensor

The overall structure of the line spot spectral confocal sensor is illustrated in Figure 2. The light source in the structure uses a fiber with a diameter of 50 μm, enabling the white light source to function as a point-light source after passing through the fiber. A slit is not directly used to generate a linear light source because it significantly reduces light intensity, preventing sufficient signal from returning to the spectrometer. To improve the light source utilization, a double-glued lens and a cylindrical lens were employed to convert the point light source into a line-light source. This method effectively enhances the light intensity and improves the object side of the numerical aperture (NA). In the dispersion section, a binary diffractive lens is used to replace the traditional refractive lens group, eliminating aberrations caused by light passing through spherical lenses and reducing the overall design complexity. The focusing section employs a 20× microscope objective, which improves the system’s measurement accuracy and axial resolution and effectively minimizes the impact of the binary diffractive lens.

## 3. Simulation Analysis

### 3.1. Simulation Analysis of Binary Diffractive Lens

A binary diffractive lens is the core component of the system, and its dispersion characteristics significantly affect the overall system performance. The design can be optimized to meet the system requirements through simulation analysis of its structure and optical properties. Based on the overall structural design requirements, the design parameters of the binary diffractive lens must satisfy the requirements listed in Table 1.

When the material influence is neglected, a Taylor expansion of the equation for the variation in the diffractive lens focal length with wavelength yields is shown in Equation (1).(1)$$f\left(\lambda \right)=\frac{f\left(\lambda_{c}\right)\lambda_{c}}{\lambda }={\sum_{i=0}^{n}\left(-1\right)}^{n}f\left(\lambda_{c}\right){\left(\frac{\lambda -\lambda_{c}}{\lambda_{c}}\right)}^{n}$$

In the above equation, λ denotes wavelength, $\lambda_{c}$ represents the center wavelength of the design, $f(\lambda_{c})$ denotes the focal length corresponding to $\lambda_{c}$.

To conduct a specific analysis of the binary diffraction lens, the effective focal length (EFFL) operand is used in ZEMAX (2023 R1) to control the focal length and set the P^2^, P^4^, and P^6^ coefficients of the binary surfaces as variables. These coefficients are optimized by the wavefront root mean square (RMS) evaluation function to satisfy the design requirements. Some simulation data are obtained as shown in Table 2:

Figure 3a,b illustrates that the primary aberration generated by a single diffractive lens during focusing is a spherical aberration. The axial chromatic aberration curves for different wavelengths show that the spherical aberration reaches the millimeter scale. Consequently, using a single diffractive lens for focusing results in a large spot size increases the measurement error and reduces the axial resolution. Therefore, it is generally necessary to match other lenses. However, the axial chromatic aberration curves also reveal that the spherical aberration curves for different wavelengths exhibit almost identical trends, indicating consistent spherical aberration characteristics of various wavelengths. This consistency facilitates a unified correction across the entire spectral band. The chromatic focal shift curves reveal that the single diffractive lens has a wide operating range and relatively good dispersion performance, achieving 29.467 mm over a wavelength range of 400–700 nm, with an R^2^ of 0.9790.

### 3.2. Microscope Objective Simulation Analysis

To address the limitations of using binary diffraction lenses, measuring the range of the control system, and improve the measurement accuracy and resolution, we integrate a 20× microscope objective lens behind the binary lens to magnify the measured area. Owing to the small aberration and outstanding image quality of the objective lens, it introduces negligible additional aberrations, effectively mitigating issues caused by using binary lenses. As shown in Figure 4, the imaging of the full field of view of the used micro-objective is evaluated by analyzing the spot diagram and the modulation transfer function (MTF) for four wavelengths between 480 and 650 nm. The simulation results indicate that the spot diagram RMS radius is smaller than the airy spot radius in the whole wavelength range, reaching the diffraction limit. Additionally, the Nyquist cutoff frequency is greater than 0.7 at 100 lp/mm, indicating the outstanding imaging effect of the microscope objective.

### 3.3. Overall Structure Simulation Analysis

By performing a simulation of the aforementioned overall structure, using the RMS evaluation function and operands to control the structural parameters of the overall system, the linearity of the chromatic focus shift curve, the size of the line spot, and the full width at half-maximum (FWHM) of the Point Spread Function (PSF) are optimized. The optimized chromatic focus shift curve is obtained as shown in Figure 5a. The dispersion range is 825 μm, and the R^2^ = 0.9983, which is excellent linearity. The out-of-focus line diagram for various wavelengths is shown in Figure 5b. It can be observed that the wavelengths 480 nm, 550 nm, and 650 nm focus at different positions within the range of −450–375 μm, exhibiting significant defocus. The length and width of the light spot were approximately 3.8 mm, and the width was approximately 0.2 mm. Huygens PSF analysis of the line spot on the image plane obtained using ZEMAX (2023 R1) is shown in Figure 6a. The transverse and longitudinal cross-section distributions of the PSF of the signal received by the spectrometer are shown in Figure 6b and Figure 6c, respectively. The results indicate that the use of line spot signal superposition does not affect the bandwidth of the return signal spot diffusion function.

## 4. Experiment

To analyze the stability difference between line spot spectral confocal sensors and point spot spectral confocal sensors, this study constructed a set of line spot and point spot confocal sensors using the same measurement range, diffractive lens, and working wavelength band. We also used the same operation procedure to perform system calibration and stability comparison experiments for these two sets of sensors.

### 4.1. System Calibration Experiment

System calibration experiments are fundamental for accurate displacement and thickness measurements, as they provide the calibration curves necessary for subsequent measurements. The displacement stage chosen is a high-precision, large-stroke linear piezoelectric stage, model PPS-20, with a closed-loop control resolution of up to 2 nm. As shown in Figure 7, the experimental setup was placed on a marble platform, with a silicon wafer attached to the nano-positioning stage to measure the sample. During the calibration experiment, the total calibration range was set to 800 μm, with the nano-positioning stage moving axially by 20 μm increments. After each movement, the spectrometer was used to collect data ten times, and this process was repeated until the entire range of motion was covered. The average value of ten data points collected at each position was used as the measurement value for the corresponding point. Forty-one measurement points were obtained after the entire movement. Polynomial fitting was performed on these data using MATLAB (R2024a). Fitting all the data using a single expression can only control the fitting error to about 0.12 μm even if the ninth-order expression is used, so the segmental fitting is used to control the fitting error of all the data to be within 0.02 μm.

The calibration curves and some interval fitting errors are illustrated in Figure 8a with an R^2^ = 0.9954 and outstanding linearity. The calibrated curves were used for displacement measurements. The same silicon wafer, fixed on the nano-displacement stage, was used as the measured object. Three measurements were conducted for the full-scale range, and the data of the nano-displacement stage and sensor values were recorded once for every 40 μm of movement, resulting in a total of 20 points. The error curves in Figure 8b indicated a maximum error of 0.037 μm across the three measurements, with a maximum standard deviation of 0.018 μm.

The same procedure was repeated for the point spot sensor. The calibration and error curves are shown in Figure 9a,b, where the correlation coefficient of the calibration curve was 0.9951. The maximum error across the three displacement measurements was 0.065 μm, and the highest standard deviation was 0.035 μm.

### 4.2. Axial Resolution Experiment

The experiment used a silicon wafer as the target, with the nano-displacement stage set to move in steps of 20 nm, 30 nm, 40 nm, and 50 nm. Spectral data were collected 1500 times after each movement, and the steps plotted based on the measured data are shown in Figure 10. The results indicated that at a displacement stage step distance of 20 nm, the step edges were more pronounced, and the difference between the mean value of the two neighboring steps and the theoretical value was larger. However, when the step distance was ≥30 nm, the steps became distinct, and the step uniformity improved. Therefore, the system’s axial resolution was better than 30 nm.

### 4.3. Static Stability Experiment

Fixing a gauge block on a nano displacement stage and adjusting its state ensured the spectral signal detected by the line spot sensor was located within the measurement range. Subsequently, the nano displacement stage was disconnected to eliminate the effect of its vibration. A spectrometer was used to collect four sets of data, each comprising 10,000 data points. The data for the last three sets were subtracted from the results of the first set to obtain three sets of relative deviation data in the static situation. The same procedure was repeated with the point light-spot sensor. The difference in static stability between the two sensors is shown in Figure 11a. The results obtained by plotting the frequency histograms of the six sets of data and Gaussian fitting are shown in Figure 11b,c. The standard deviation of static stability of the line spot sensor is 0.011 μm, and that of the point spot sensor is 0.018 μm. From the experimental results, the standard deviation of the static stability of the two sensors does not differ much and is relatively small, indicating that the confocal sensor is not significantly affected by external environmental disturbances, and the static stability of the line spot sensor is slightly better than that of the point spot sensor.

### 4.4. Dynamic Stability Experiment

The dynamic stability experiment was conducted by maintaining the same conditions as the static stability experiment, ensuring continuous signal capture by the sensor as the nano-positioning stage moved. The nano-positioning stage was controlled to move unidirectionally along the longitudinal axis for 4 mm, with the entire process recorded using a spectrometer. A total of 10,000 data points were collected, and the experiment was repeated four times for each type of sensor. To calculate the relative deviation for three motion cases, the values from the last three measurements were subtracted from the values of the first measurements by starting point alignment. Outliers were removed before plotting frequency histograms, followed by Gaussian fitting. The difference in dynamic stability between the two sensors is shown in Figure 12a. The measurement results showed that the maximum deviation for the line spot sensor was 0.053 μm, with a standard deviation of 0.013 μm, while the maximum deviation of the point spot sensor was 0.315 μm, with a standard deviation of 0.064 μm (Figure 12b,c). In the scanning measurement process, the most direct interference is the vibration and angular pendulum of the measured object in the process of movement. From the experimental results, it can be observed that the point spot sensor often has several-hundred-nanometer jumps during the continuous scanning measurement process, while the line light spot sensor is relatively smooth throughout the scanning process. It is also found that the static stability standard deviation and dynamic stability standard deviation of the line spot sensor only differ by a few nanometers, while the difference in the point spot sensor is tens of nanometers. It is evident that the line spot sensor can obviously eliminate the influence of vibration and angular pendulum and other disturbances during the continuous scanning measurement process, and improve the stability of the sensor’s dynamic measurement.

### 4.5. Standard Step Scanning Experiment

The sensor is fixed vertically on the liftable air-bearing guide rail, and then the sample with four standard steps of 300 μm, 100 μm, 50 μm, and 20 μm is fixed on the two-dimensional motion platform, and the distance between them is adjusted so that it can complete the scanning of the whole step sample. The relative position relationship between the scanning spot and the step and the scanning process are shown in Figure 13, which controls the two-dimensional motion platform to reciprocate in the X direction at a speed of 5 mm/s, straightness 0.015 μm, and flatness 0.020 μm in the X direction of the platform. A total of five sets of scans are performed to obtain ten step scanning results. An average of 80 data points were measured by each step sensor, and their average value was calculated as the current step measurement. The error curves of the ten scan results are shown in Figure 14, where the maximum value is −0.047 μm, and the standard deviation of the ten measurement errors for each step is 0.011 μm at maximum. From the experimental results, it can be concluded that the line light spot sensor has better accuracy and stability during the scanning process.

## 5. Conclusions

In this study, a line spot spectral confocal displacement sensor was designed to improve stability during surface profile measurements. A binary diffractive lens was used in the dispersion module, simplifying the overall design compared with using a refractive lens. The addition of a cylindrical mirror and a 20× microscope objective lens to achieve a large spot and reduce the use of binary lenses introduced by the millimeter-level spherical aberration, improving the sensor measurement accuracy and stability. In terms of signal extraction, by superposition of all the signals scanned by the line spot and then taking its peak wavelength, according to the extracted peak wavelength to analyze the depth information of the surface of the object to be measured, the measured value is equivalent to the surface of the object to be measured filtering process. To validate the effectiveness of the proposed solution, two sets of line spot and point spot confocal sensors with the same dispersion range and operating band were constructed using the same dispersive elements as in the control group, and the same experimental protocol was used for both sets of sensors. The two sensors were calibrated at the full range of 825 μm by means of a displacement stage, followed by accuracy, resolution, stability analysis, and standard stage scanning experiments. From the experimental results, it can be concluded that the axial resolution of the line spot sensor is better than 30 nm and the accuracy is better than that of the point confocal sensor. The static stability of the two groups of sensors does not differ much, indicating that the confocal sensors are not significantly affected by ambient noise. However, according to the experimental results of dynamic stability, the dynamic stability of the line spot is improved by 79% relative to the conventional point spot. From the standard step scanning experiment, it is evident that the measurement error of the line spot sensor can basically remain stable during the continuous scanning process, indicating that this sensor can maintain a good measurement state for a long period of time. In summary, the stability of the spectral confocal sensor based on the line spot proposed in this paper has been significantly improved.

## Figures and Tables

**Figure 1 sensors-25-01337-f001:**
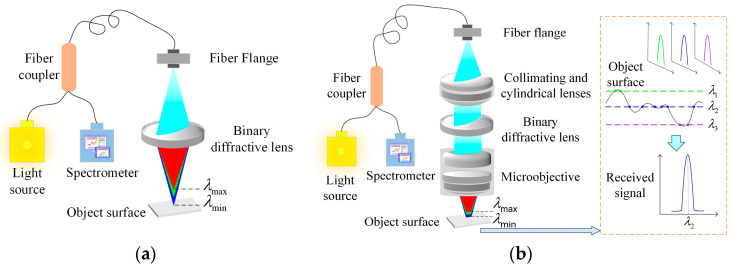
Spectral confocal sensor measurement principle: (**a**) Conventional diffractive lens point confocal sensor measurement system; (**b**) The line spot confocal sensor measurement system proposed in this paper.

**Figure 2 sensors-25-01337-f002:**
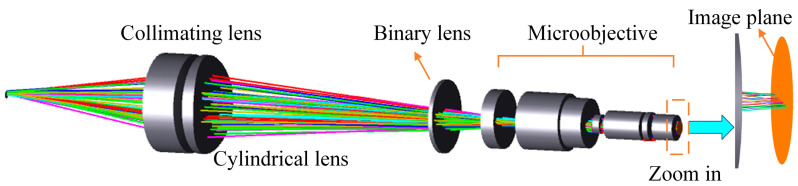
Structure of the proposed line spot spectral confocal sensor.

**Figure 3 sensors-25-01337-f003:**
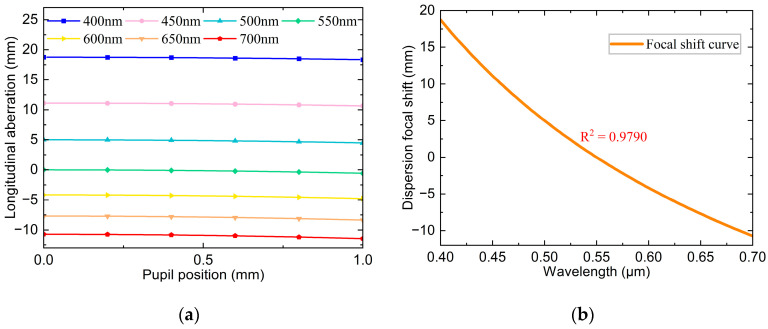
Simulation results of optimized binary diffraction lens at 400–700 nm: (**a**) Axial chromatic aberration curve; (**b**) Chromatic focal shift curve for binary diffractive lens.

**Figure 4 sensors-25-01337-f004:**
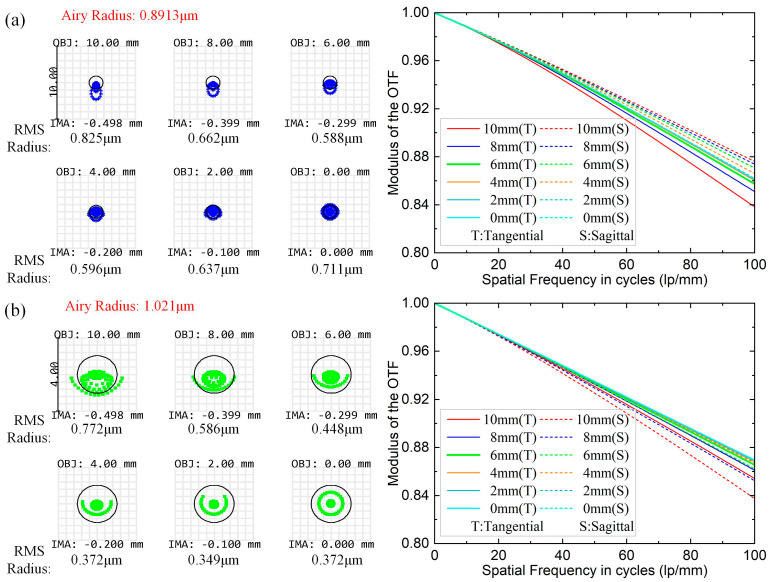
Image quality analysis of the microscope objective at four specific wavelengths: (**a**) 480 nm (blue) spot diagram and MTF curve; (**b**) 550 nm (green) spot diagram and MTF curve; (**c**) 600 nm (red) spot diagram and MTF curve; (**d**) 650 nm (yellow) spot diagram and MTF curve.

**Figure 5 sensors-25-01337-f005:**
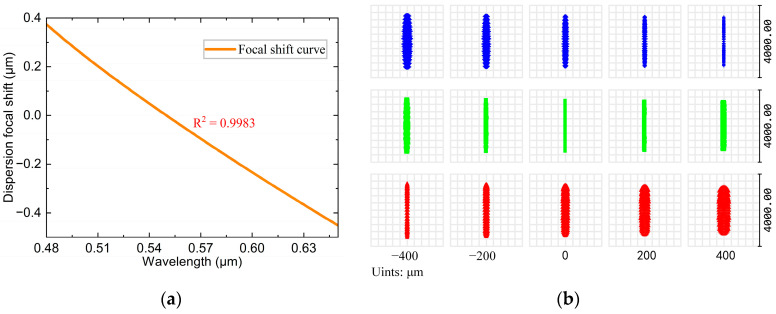
Simulation analysis of the overall structure after ZEMAX optimization: (**a**) Chromatic focus shift curve of line spot spectral confocal sensor between 480 and 650 nm; (**b**) 480nm (blue), 550 nm (green), 650 nm (red) defocusing between −400~400 μm.

**Figure 6 sensors-25-01337-f006:**
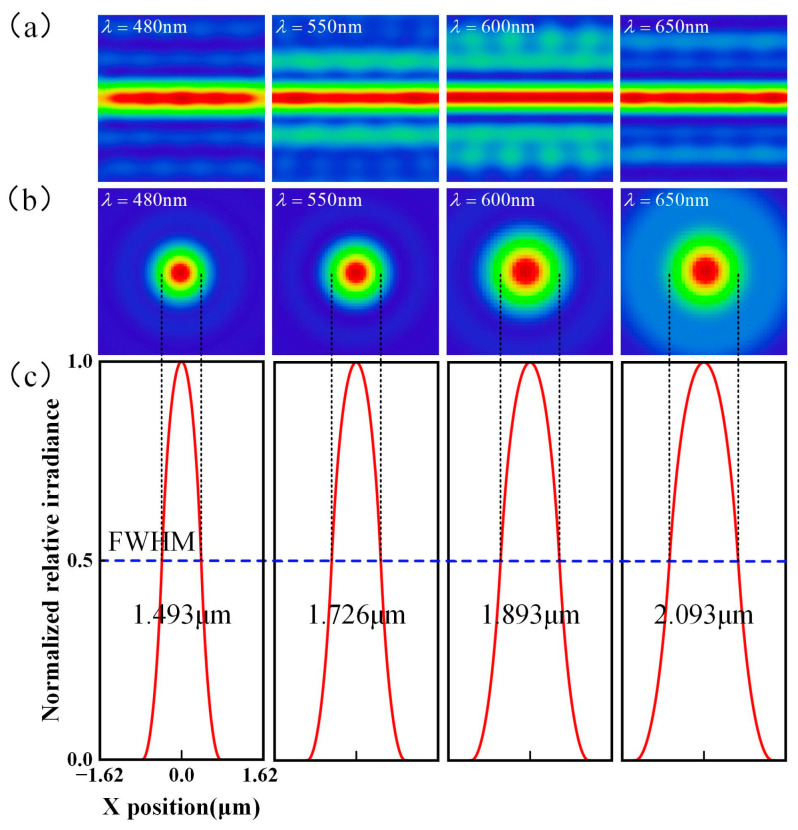
PSF analysis of the spot generated by a line spectral confocal sensor and the spot received by a spectrometer: (**a**) Image plane Huygens PSF simulation, (**b**) Return signal PSF transverse cross-section distribution, (**c**) Return signal PSF longitudinal cross-section distribution.

**Figure 7 sensors-25-01337-f007:**
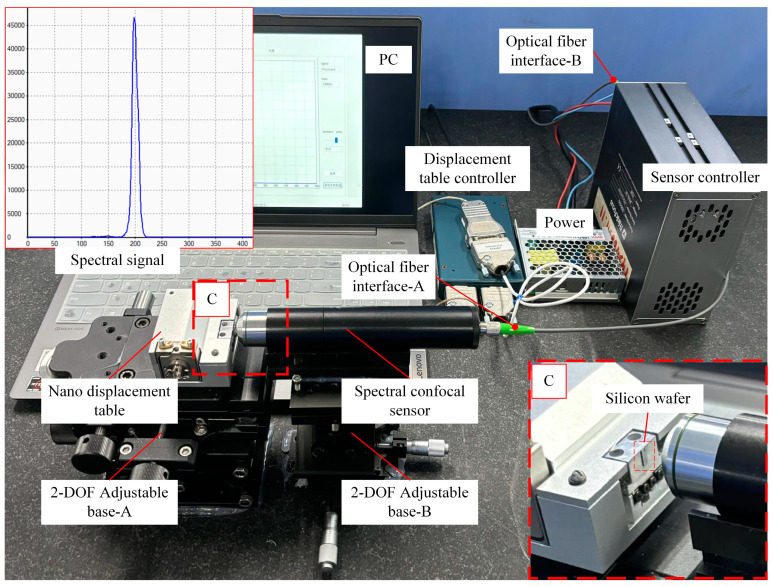
Experimental platform for calibration. A, B indicates two base adjustment brackets, and C represents a localized magnified view of the lens and the subject.

**Figure 8 sensors-25-01337-f008:**
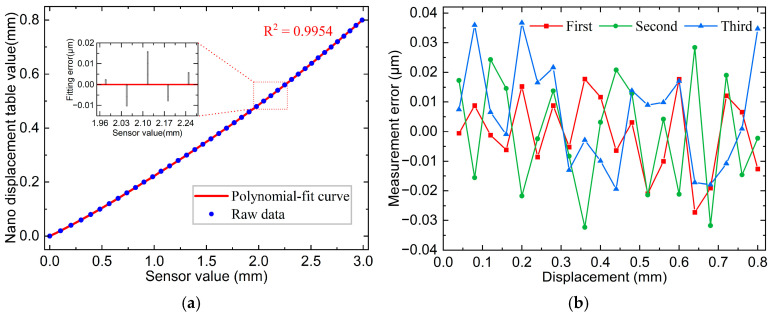
Line spot sensor calibration results and measurement errors: (**a**) Line spot 800 μm calibration curve and some interval fitting error; (**b**) Line spot full-scale measurement error curve.

**Figure 9 sensors-25-01337-f009:**
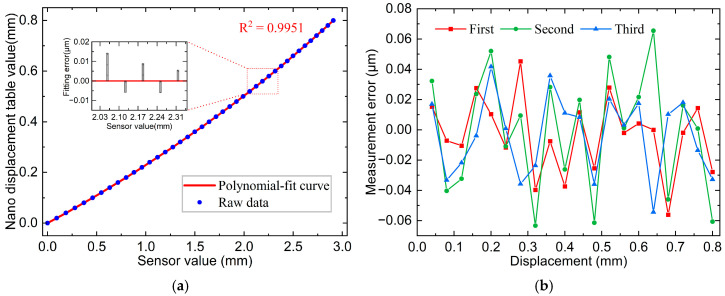
Point spot sensor calibration results and measurement errors: (**a**) Point spot 800 μm calibration curve and some interval fitting error; (**b**) Point spot full-scale measurement error curve.

**Figure 10 sensors-25-01337-f010:**
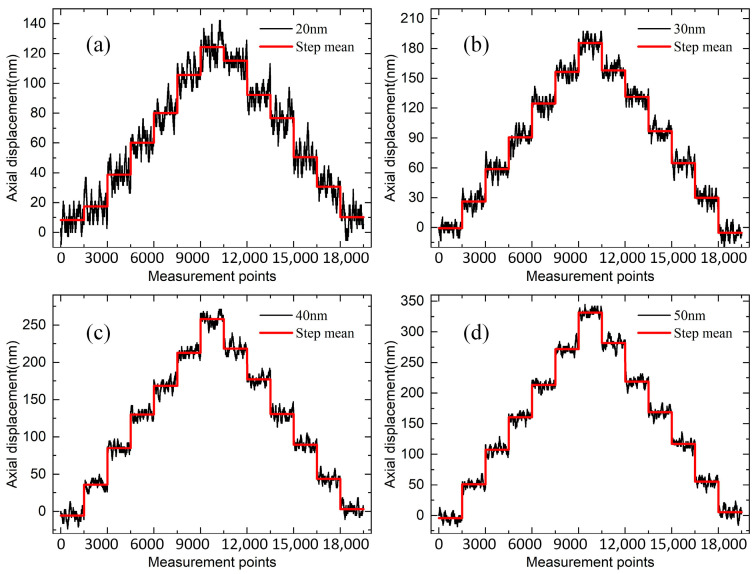
Experimental results of axial resolution for different steps: (**a**) 20 nm; (**b**) 30 nm; (**c**) 40 nm; (**d**) 50 nm.

**Figure 11 sensors-25-01337-f011:**
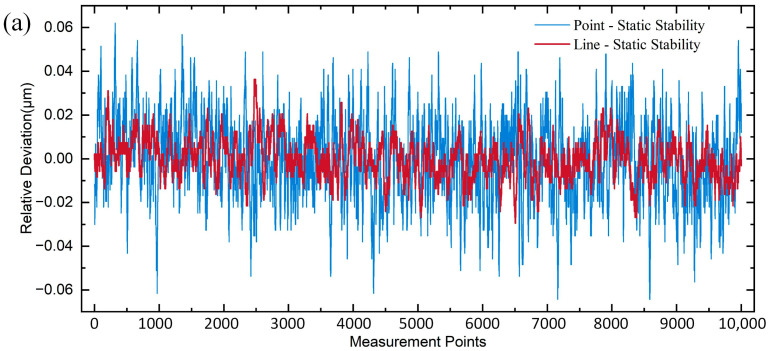
Static stability analysis: (**a**) Point-line spot static stability differences; (**b**) Point spot static stability frequency histogram and Gaussian fitting; (**c**) Line spot static stability frequency histogram and Gaussian fitting.

**Figure 12 sensors-25-01337-f012:**
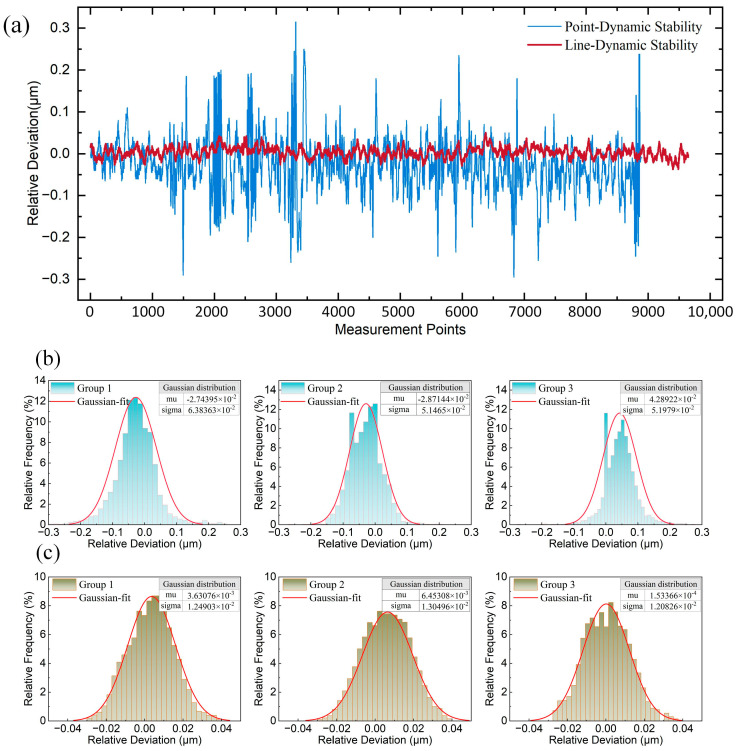
Dynamic stability analysis: (**a**) Point-Line spot dynamic stability differences; (**b**) Point spot dynamic stability frequency histogram and Gaussian fitting; (**c**) Line spot dynamic stability frequency histogram and Gaussian fitting.

**Figure 13 sensors-25-01337-f013:**
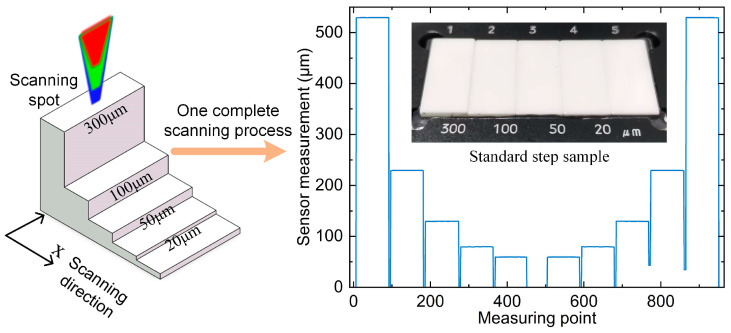
Relative position of the scanning spot to the step and one complete scanning process.

**Figure 14 sensors-25-01337-f014:**
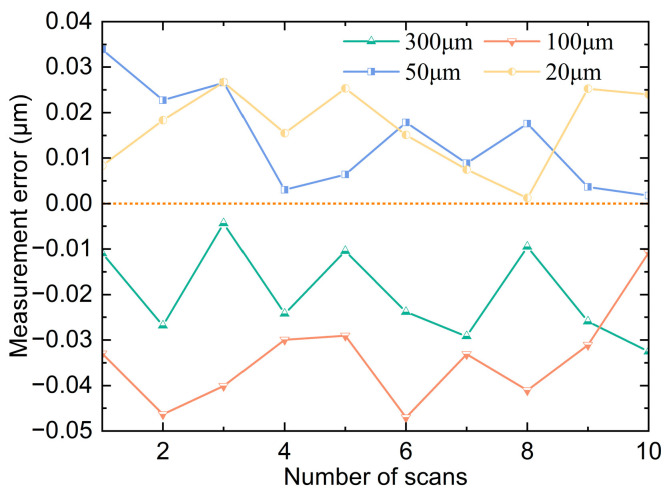
Measurement error curves for each step.

**Table 1 sensors-25-01337-t001:** Design parameters of binary diffractive lens.

Parameter	Value
Design center wavelength (nm)	550
Focal length (mm)	50
Diffraction level	−1
Thicknesses (mm)	1.2

**Table 2 sensors-25-01337-t002:** Binary diffraction lens simulation results.

Surface Type	Thickness	Material	Radius	Norm Radius	P^2^	P^4^	P^6^
Object	Inf	-	-	-	-	-	-
Standard	1.2	N-BK7	9.5	-	-	-	-
Binary	50.005	-	9.5	7.5	6425.396	−333.748	80.760

## Data Availability

Data underlying the results presented in this paper are not publicly available at this time but may be obtained from the authors upon reasonable request.

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
