# Peer review of "Anti-Interference Spectral Confocal Sensors Based on Line Spot"

_sensors, 2025, doi:10.3390/s25051337_

Round 1

Reviewer 1 Report

Comments and Suggestions for Authors

The authors present an improvement in the conditioning of the illumination beam in confocal microscopy. Instead of using a circular spot, they use a linear spot that allows to reduce the interference of the environment in the measurements.

The manuscript is well written, well structured and shows enough progress to be accepted for publication.

The only deficiency I see is that in the introduction only the last three lines (106-108) mention what the difference or progress of the manuscript would be. I suggest going into a little more depth on these differences and progress.

Also, in the Taylor expansion (Eq. 1) lambda and lambda c should be defined.

And in line 55 it is possible that “o” is “to”.

Author Response

Comment 1: The only deficiency I see is that in the introduction only the last three lines (106-108) mention what the difference or progress of the manuscript would be. I suggest going into a little more depth on these differences and progress.

Response: Thank you for pointing this out. I agree with this comment. On the third page of the article, in lines 103-114, I add three articles on aspects of stability enhancement, briefly describing their approaches to enhancing stability, and in lines 119-123 I summarize the major current research directions in stability enhancement and the approaches to enhancing stability proposed in this article.

Comment 2: Also, in the Taylor expansion (Eq. 1) lambda and lambda c should be defined.

Response: Thank you for pointing this out. I agree with this comment. Additional explanations of the parameters in the equation are provided in red font on page 5, lines 179-180.

Comment 3: And in line 55 it is possible that “o” is “to”.

Response: Thank you for pointing this out. I agree with this comment. In line 55 it should be to. Changes have been made in the text on page 2, line 55, in red.

Reviewer 2 Report

Comments and Suggestions for Authors

The paper proposes a line-spot-based measurement method using a binary diffractive lens and a cylindrical lens, and investigates its performance.  The obtained results show an improvement in stability during the scanning process.

The paper is well-written; however, there are some unclear points. Please address the following:

1.  The main issue is that there is no comparison of the proposed system to conventional ones. It is difficult for readers to judge if the proposed system is novel. 
Please compare the actual value of the stability and clarify the actual advantages of the proposed system over conventional ones.
2.  In the conclusion, please add more discussion about the above points, as the conclusion in the abstract does not match the statement in the conclusion section.
3.  What is the main improvement of the measurement system and to what is it attributed? Could it be the binary diffractive lenses?
4.  Please add a comment on the effect of the mechanical stage.
5.  Please add more information on Fig. 10. For example, Fig. 10(a) 20 nm, (b) 30 nm, (c) 40 nm, and (d) 50 nm steps.

Author Response

Comment 1: The main issue is that there is no comparison of the proposed system to conventional ones. It is difficult for readers to judge if the proposed system is novel.

Please compare the actual value of the stability and clarify the actual advantages of the proposed system over conventional ones.

Response: Thank you for pointing this out. I agree with this comment. In lines 308-312 on page 11 of the article, the experimental results for static stability are analyzed in more detail, and the actual values of stability for both are not very different, leading to the conclusion that the confocal sensor does not have a significant effect on environmental disturbances. On page 12 of the article, lines 333-342, the dynamic stability of the experimental results of a more in-depth analysis of the two sensors dynamic stability of the actual value of the difference between the two sensors is more obvious, the point of the confocal sensor throughout the process of the obvious jump, while the line confocal sensor throughout the process of the relatively smooth, with the standard deviation of their stability to calculate the stability of the line spot compared to the point of the spot to enhance the stability of the 79%.

Comment 2: In the conclusion, please add more discussion about the above points, as the conclusion in the abstract does not match the statement in the conclusion section.

Response: Thank you for pointing this out. I agree with this comment. Firstly, I have corrected the inconsistency between the abstract and the conclusion in the conclusion section of the article, lines 371-373 on page 14, due to a logical error in my narrative. And I added more analysis of the experimental results in lines 381-393, so that the conclusion of the article that the stability is improved is more strongly supported.

Comment 3: What is the main improvement of the measurement system and to what is it attributed? Could it be the binary diffractive lenses?

Response: Thank you for pointing this out. I agree with this comment. The main improvement of the system is the addition of cylindrical mirrors and microscopic objectives to the traditional point spot to realize the effect of large spot, in order to achieve the purpose of improving the stability of the sensor. The binary diffractive lens is used in the paper because of its very good dispersion ability, which can simplify the overall structure to a great extent and have better effect compared with the refractive lens set. Additional clarification of this issue is provided in lines 137-139 on the third page of the text.

Comment 4: Please add a comment on the effect of the mechanical stage.

Response: Thank you for pointing this out. I agree with this comment. Add the model number and key parameters of the displacement table to lines 253-254 on page 8 of the article. Added the straightness and flatness of the two-dimensional motion stage in the X-direction on line 355 on page 13 of the article.

Comment 5: Please add more information on Fig. 10. For example, Fig. 10(a) 20 nm, (b) 30 nm, (c) 40 nm, and (d) 50 nm steps.

Response: Thank you for pointing this out. I agree with this comment. I have added the numbers a, b, c, and d to Figure 10 on page 10, line 294 of the text. And supplemented the title of the figure on lines 295-296.

Figure 10. Experimental results of axial resolution for different steps: (a) 20 nm; (b) 30 nm; (c) 40 nm; (d) 50 nm.